# Role of the endothelial cell apolipoprotein E receptor 2 in modulating the effects of apoE3 and apoE4 on insulin blood-brain barrier transport

Peter Thomas[1], Van Nguyen[1], Riley Weaver[1], Kim Hansen[1], Anastasia Sacharidou[2], William A. Banks [1,3], Chieko Mineo[2,4], Philip W. Shaul [2], Elizabeth M. Rhea [1,3]*

1 Geriatric Research Education and Clinical Center, Veterans Affairs Puget Sound Health Care System, Seattle, Washington, United States of America, 2 Center for Pulmonary and Vascular Biology, Department of Pediatrics, University of Texas Southwestern Medical Center, Dallas, Texas, United States of America, 3 Department of Medicine, Division of Gerontology and Geriatric Medicine, University of Washington, Seattle, Washington, United States of America, 4 Department of Cell Biology, University of Texas Southwestern Medical Center, Dallas, Texas, United States of America

* meredime@uw.edu

## Abstract

Apolipoprotein E receptor 2 (apoER2), a primary receptor for apoE, has recently been linked to Alzheimer's disease. Compared with the most common form of apoE, apoE3, the apoE4 isoform increases the risk for developing Alzheimer's disease. ApoE4 impairs brain insulin signaling, a feature of Alzheimer's disease that correlates with cognitive decline. Insulin availability in the brain largely depends on blood-brain barrier (BBB) transport and contributes to brain insulin signaling. We have previously shown that the apoE4 isoform leads to regional reductions in insulin BBB transport in mice on a Western diet compared to apoE3 isoform. However, how insulin transport across the BBB is regulated by apoE isoforms is not well understood. Here we investigated a role of endothelial apoER2 in the effects of apoE isoforms on insulin BBB transport, using mice genetically expressing human apoE3 or apoE4 and expressing or lacking endothelial apoER2. We found that a loss of endothelial apoER2 did not overtly affect insulin BBB transport in either apoE3- or apoE4-expressing mice, except in the frontal cortex and pons/medulla, where decreased transport was observed in apoE3 mice lacking endothelial apoER2. These findings indicate that the effect of apoE4 on insulin BBB transport is largely independent of endothelial apoER2. In contrast, endothelial apoER2 may regulate insulin BBB transport in limited regions of the brain through its binding to apoE3.

## Introduction

The apolipoprotein E receptor 2 (apoER2) is a primary receptor for apolipoprotein E (apoE), as well as the glycoprotein Reelin, and it is a member of the low-density lipoprotein (LDL) receptor family, encoded by the *LRP8* gene. ApoER2 expression

**Data availability statement:** All relevant data are within the paper and its Supporting Information files.

**Funding:** This work was supported by both the Veterans Affairs (general R&D funding – no specific grant) and the National Institute of Health (specific grants listed). The Veterans Affairs Research and Development (EMR and WAB), NIH R01HL144969 (PWS), R01HL144969-S1 (PWS), and NIH R01HD094395 (CM) helped support this work The funders had no role in study design, data collection and analysis, decision to publish, or preparation of the manuscript. We do not have anything else to disclose.

**Competing interests:** The authors have declared that no competing interests exist.

is enriched in the brain [1] and is linked to Alzheimer's disease (AD) through gene polymorphism studies [2] and its intracellular trafficking changes due to mutations in AD related genes [3]. Additionally, individuals with the apoE4 allele, compared to the more common apoE3 allele, have an increased risk for developing AD and cognitive decline [4]. ApoER2 interacts with apoE to stimulate amyloid-β production [5,6]. ApoER2 is most abundantly expressed in mouse brain endothelial cells compared to other brain cell types [7] and it has been targeted for delivery of therapeutics to the brain [8].

ApoER2 at the blood-brain barrier (BBB) regulates leukocyte adhesion and entry into the brain [9], highlighting an important role of this receptor at the BBB. ApoE3-apoER2 signaling in the aortic endothelium stimulates endothelial NO synthase (eNOS), which is localized to caveolae, aids in endothelial repair, and activates anti-inflammatory events [10,11]. These effects of apoE3 are antagonized by apoE4 [11].

Insulin availability in the brain is primarily supplied from circulation through transport across the BBB [12]. Insulin BBB transport is reduced in obesity [13,14] and in mice fed a high fat diet [15]. Brain insulin action is impaired in AD and brain insulin resistance negatively correlates with cognition [16]. Our prior research indicates that insulin BBB transport may be mediated by caveolae [17]. We and others have shown that insulin BBB transport can occur independent of the insulin receptor [18–20], leaving the BBB transporter for insulin currently unknown.

We have previously shown that the apoE4 isoform leads to regional reductions in insulin BBB transport in male mice on a Western diet compared to apoE3 isoform [15]. Others have shown that apoE4 reduces brain insulin signaling [21,22]. Identifying ways to optimize BBB transport of insulin, particularly in apoE4 carriers, may prevent impairments in brain insulin signaling, slowing the rate of cognitive decline. For that reason, the present project aimed to determine the role of endothelial apoER2 in the impacts of apoE3 and apoE4 on insulin BBB transport.

## Materials and methods

### Animals

Mice in which the murine apoE gene has been replaced by human apoE3 or apoE4 originally developed by Dr. Maeda [23] were bred in house with floxed apoER2 mice (apoER2$^{fl/fl}$) [24] and mice expressing Cre recombinase under the control of the VE-cadherin promoter [25] to mediate endothelial cell-specific gene silencing of apoER2 (apoER2$^{\Delta EC}$) [24]. Effective deletion of apoER2 in endothelium in apoER2$^{\Delta EC}$ has previously been shown in two independent projects, along with demonstration that there is no change in apoER2 expression in bone marrow-derived myeloid cells [24,26]. Male mice in the following four groups were studied: apoE3;apoER2$^{fl/fl}$, apoE3;apoER2$^{\Delta EC}$, apoE4;apoER2$^{fl/fl}$, and apoE4;apoER2$^{\Delta EC}$ mice. The mice had *ad libitum* access to food and water and were kept on a 12/12 hour light/dark cycle. They were fed a high-fat, high cholesterol (HF/HC) diet (TD88137, 21% fat and 0.2% cholesterol) [27] beginning at 5 weeks of age for 8 weeks. The mice were then fasted

overnight and insulin BBB pharmacokinetics were assessed. All procedures complied with the National Institutes of Health Guide for the Care and Use of Laboratory Animals and with IACUC approval at both UTSW and the VA Puget Sound. The mice were bred, and the study was performed at facilities approved by the Association for Assessment and Accreditation of Laboratory Animal Care International (AAALAC).

## Radioactive labeling of insulin and albumin

Insulin and albumin were radioactively labeled as previously described [15]. Briefly, human insulin (10 μg, Sigma-Aldrich, St. Louis, MO, USA) was radioactivity labeled with 1 mCi Na$^{125}$I (Perkin Elmer, Waltham, MA, USA) using chloramine-T (Sigma-Aldrich). Bovine serum albumin (BSA, Sigma-Aldrich) was radioactively labelled with $^{99m}$Tc (GE Healthcare, Seattle, WA, USA). Radioactively labeled insulin ($^{125}$I-insulin) and radioactively labeled albumin ($^{99m}$Tc-albumin) were purified on a Sephadex G-10 (Sigma-Aldrich) column and quality-controlled steps were taken, including acid precipitation, with 30% trichloroacetic acid (TCA, Sigma-Aldrich), to validate intact radioactively labeled substrates.

## BBB pharmacokinetic transport of $^{125}$I-insulin

After 8 weeks on the HF/HC diet, mice were anesthetized with 40% urethane (0.15 ml intraperitoneal injection) to minimize pain and distress during the terminal study. Tracer amounts of $^{125}$I-insulin (1x10$^6$ cpm) and $^{99m}$Tc-albumin (5x10$^5$ cpm) circulated for 0.5–4 min following an intravenous injection in 1% BSA/lactated Ringer's solution (LR) (0.2 ml). The $^{99m}$Tc-albumin was co-injected as a marker for vascular space [28] and cannot be used in this study as a measurement of BBB integrity due to the short circulation time and lack of brain vascular washout. Blood from the left carotid artery was collected from individual mice at each time point. Afterwards, mice were decapitated and the whole brains quickly removed, dissected into regions according to the method described in [29], and weighed. The arterial blood was centrifuged at 5,400$g$ for 10 min and serum was collected. The levels of radioactivity in serum (50 μl) and brain regions were counted in a gamma counter (Wizard2, Perkin Elmer, Waltham, MA). The brain/serum (B/S) ratios were graphically displayed against their respective exposure times (Expt). Expt was calculated from the formula:

$$Exposure\ time = \frac{\int_0^t Cp(t)dt}{Cp(t)}$$

(1)

where $Cp$ is the level of radioactivity (cpm) in serum at time ($t$). Expt corrects for the clearance of tracer from the blood. The influx of insulin was calculated by multiple-time regression analysis as described by Patlak, Blasberg, and Fenstermacher [28,30]:

$$\frac{Am}{Cpt} = Ki\ (\frac{\int_0^t Cp(t)dt}{Cp(t)}) + Vi$$

(2)

where $Am$ is level of radioactivity (cpm) per g of brain tissue at time $t$, $Cpt$ is the level of radioactivity (cpm) per ml arterial serum at time $t$, K$_i$ (μl/g-min) is the steady-state rate of unidirectional solute influx from blood to brain, and V$_i$ (μl/g) is the level of rapid and reversible binding for brain which usually is a combination of vascular space plus any brain endothelial cell receptor binding. The brain/serum (B/S) ratios for insulin were corrected for vascular space by subtracting the corresponding ratio for albumin, yielding a delta B/S ratio. The linear portion of the relation between the delta B/S ratio versus Expt was used to calculate the K$_i$ (μl/g-min) with its standard error term, and the y-intercept determined as representation of the V$_i$ (μl/g) [28]. As the B/S ratio for $^{99m}$Tc-albumin did not differ across the time points for all regions except the thalamus in the apoE4;apoER2$^{fl/fl}$ group, the average vascular space for each genotype was also calculated by collapsing values across time points for each brain region.

## Statistics

Regression analyses and other statistical analyses were performed with the Prism 10 software package (GraphPad Software Inc., San Diego, CA, USA). For pharmacokinetic studies, the slope of the linear regression lines ($K_i$), reported with their correlation coefficients ($r$), and y-intercepts ($V_i$) were compared statistically using the Prism software package, as described [31]. Comparative analysis was subsequently performed for those regions demonstrated to have transport. Either a two-way ANOVA with a Fisher's LSD post hoc test was used to determine differences due to apoE isoform or endothelial apoER2 or a student's t test. Outliers were removed by the ROUT method ($Q = 1\%$); final 'n' for each group are reported in figure legends. In cases where outliers were removed from each region, this was most often from the same animal. Only p values $\le 0.05$ were considered significant.

## Results

### Brain region weights

There was no difference in whole brain weight due to loss of endothelial cell apoER2, and there were minor differences in brain region weights. There was a significant increase in frontal cortex weight in apoE4 mice ($p = 0.0014$) (S1C in S1 Fig). There was a significant decrease in the weight of the pons/medulla due to loss of endothelial apoER2 in apoE4 mice ($p = 0.017$) (S1L in S1 Fig).

### Vascular space

As described in the methods, mice received an intravenous injection of the vascular marker $^{99m}$Tc-albumin. As expected, there was no change in $^{99m}$Tc-albumin levels over time (S2 Fig) in whole brain or any of the brain regions, indicating no measurable leakage of $^{99m}$Tc-albumin during this brief circulation period, except for the apoE4;apoER2$^{\Delta EC}$ group in the thalamus ($r = 0.47$, $p = 0.026$) (S2G in S2 Fig). $^{99m}$Tc-albumin data without significant uptake over time, which included all groups except for the apoE4;apoER2$^{\Delta EC}$ group in the thalamus, was collapsed over time to increase statistical reliability and used as a marker of the vascular space for each region and $^{99m}$Tc-albumin values (Fig 1). In apoE3 mice, there were no significant differences between apoER2$^{fl/fl}$ and apoER2$^{\Delta EC}$ (Fig 1). In apoE4 mice, there was a significant increase in vascular space due to loss of endothelial apoER2 in the parietal cortex ($p = 0.007$) (Fig 1H).

### Insulin BBB pharmacokinetics

Serum clearance of $^{125}$I-insulin was not affected by the presence or absence of endothelial cell apoER2 (Fig 2). Rates of clearance and the half-life for $^{125}$I-insulin are listed in Table 1.

Linear regression analysis was used to calculate the serum clearance rate for $^{125}$I-insulin. The inverse of the slope was multiplied by 0.301 (log10) to calculate the half-time clearance. 'r' reflects the correlation coefficient and the p value refers to the significance of the linear regression.

After accounting for the amount of $^{125}$I-insulin in each serum sample, the level of vascular space, and the weight of each brain region, we quantified the unidirectional influx rate ($K_i$) of $^{125}$I-insulin across the BBB. Full transport curves are presented in S3 Fig and the calculated rates of transport ($K_i$) are presented in Fig 3 and Table 2. There was measurable transport into the whole brain in the four genotype groups (S3A Fig). There was no difference in whole brain transport within either apoE3 or apoE4 mice due to loss of endothelial apoER2 (Fig 3A, Table 2). The transport in brain regions was significantly decreased in apoE4 versus apoE3 mice in the olfactory bulb (Fig 3B) and increased in the midbrain (Fig 3K). In apoE3 mice, $^{125}$I-insulin transport was detected within each brain region (Fig 3B-3L). There was a significant decrease in the transport rate in apoE4 versus apoE3 mice in apoER2$^{fl/fl}$ mice in the frontal cortex (Fig 3C), parietal cortex (Fig 3H), and the pons/medulla (Fig 3L). There was a significant decrease in the transport rate in apoE4 versus apoE3 mice in

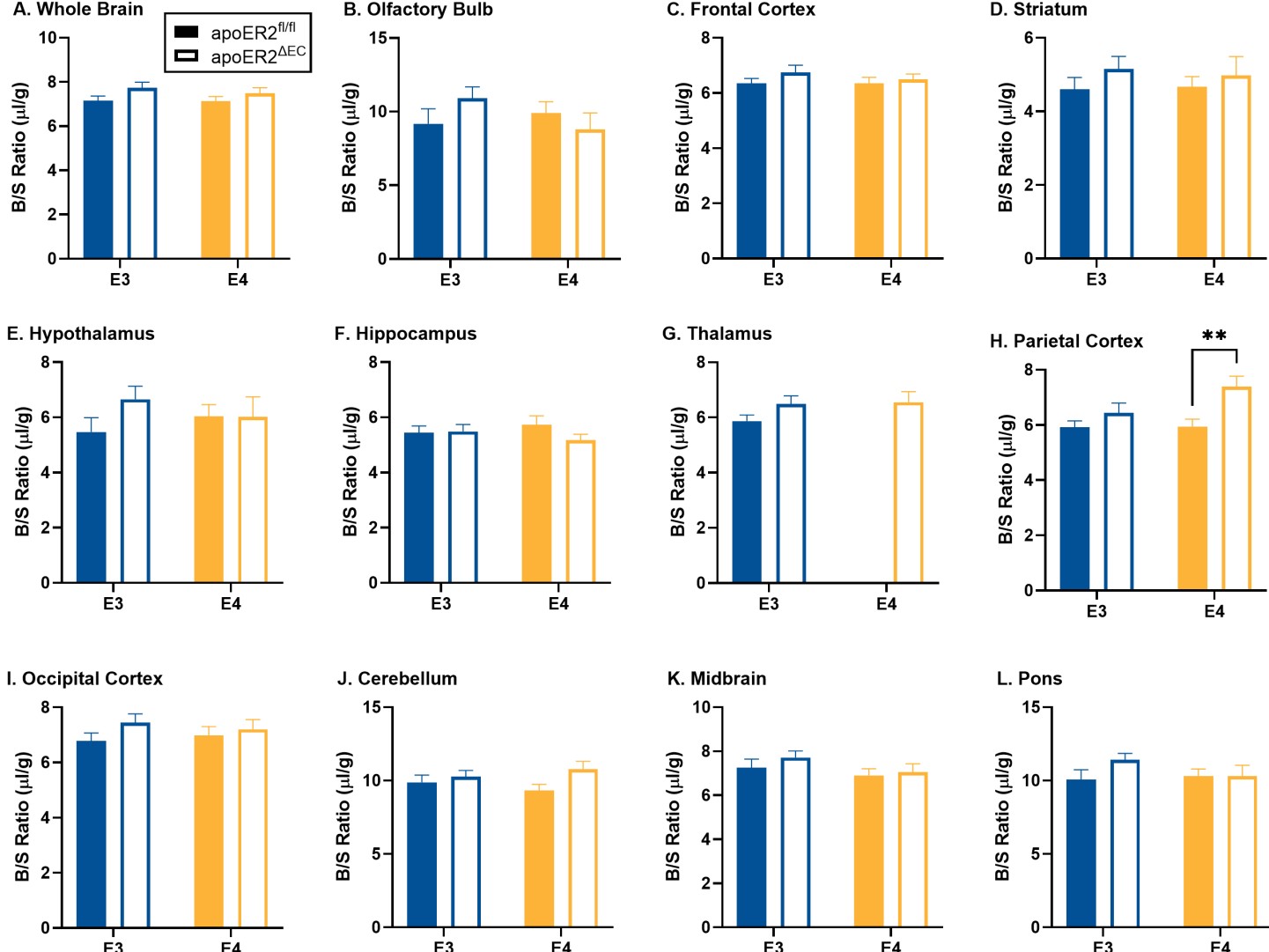

**Fig 1. Brain regional vascular space.** The effects of apoE isoform and endothelial apoER2 on vascular space are graphed for A) whole brain, B) olfactory bulb, C) frontal cortex, D) striatum, E) hypothalamus, F) hippocampus, G) thalamus, H) parietal cortex, I) occipital cortex, J) cerebellum, K) midbrain, and L) pons/medulla. Means reported ± SEM. ANOVA differences are indicated by brackets between E3 vs E4. Fisher's LSD post hoc differences are marked, *p < 0.05, **p < 0.01. Final sample sizes reflect apoE3;apoER2$^{fl/fl}$ n = 17, apoE3;apoER2$^{\Delta EC}$ n = 25, apoE4;apoER2$^{fl/fl}$ n = 24, apoE4;apoER2$^{\Delta EC}$ n = 11 for most regions with additional outliers removed by the ROUT method (Q = 1%) including, Frontal Cortex: n = 1 apoE3;apoER2$^{\Delta EC}$ (n = 24 total), Thalamus: n = 1 apoE3;apoER2$^{\Delta EC}$ n = 24 total), and n = 2 apoE4;apoER2$^{fl/fl}$ (n = 22 total), Occipital Cortex: n = 2 apoE3;apoER2$^{\Delta EC}$ (n = 23 total) and n = 1 apoE4;apoER2$^{fl/fl}$ (n = 10 total), Pons: n = 1 apoE3;apoER2$^{fl/fl}$ (n = 16 total).

apoER2$^{\Delta EC}$ mice in the occipital cortex (Fig 3I). There was an increase in the transport rate in apoE4 versus apoE3 mice in apoER2$^{\Delta EC}$ mice in the midbrain (Fig 3K).

There was a significant decrease in the transport rate in the frontal cortex (Fig 3C) and pons/medulla (Fig 3L) due to loss of endothelial apoER2 in apoE3 mice. There was no effect of endothelial apoER2 on $^{125}$I-insulin transport in apoE4 mice. In apoE4 mice, $^{125}$I-insulin transport could not be demonstrated in the striatum in either apoER2$^{fl/fl}$ or apoER2$^{\Delta EC}$ mice (Fig 3D), nor in the parietal cortex of apoER2$^{\Delta EC}$ mice (Fig 3H). The large variance in the parietal cortex precluded further comparative analysis between apoE3:apoER2$^{\Delta EC}$ and apoE4:apoER2$^{\Delta EC}$ mice. Correlation coefficients and sample

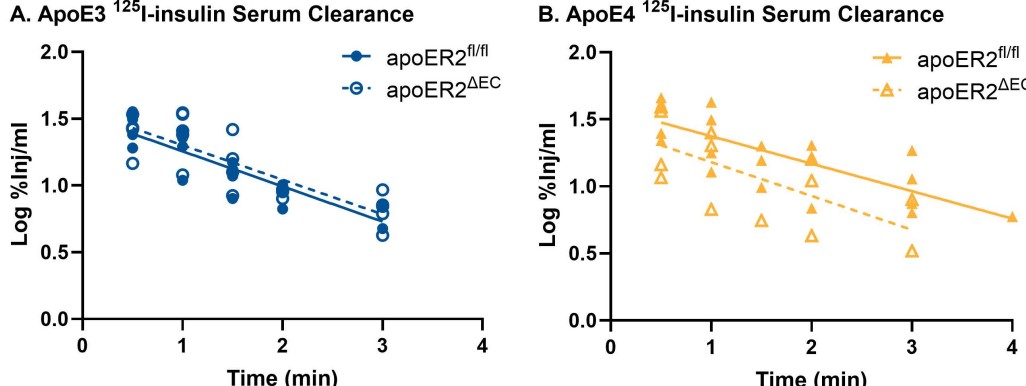

**Fig 2. $^{125}$I-insulin serum clearance within each apoE genotype was compared between apoER2$^{fl/fl}$ and apoER2$^{\Delta EC}$ mice.** There were no differences in the serum clearance rate for $^{125}$I-insulin due loss of endothelial apoER2. apoE3;apoER2$^{fl/fl}$ n = 18; apoE3;apoER2$^{\Delta EC}$ n = 23; apoE4;apoER2$^{fl/fl}$ n = 22; apoE4;apoER2$^{\Delta EC}$ n = 13.

**Table 1. Serum clearance rates.**

| Group | Half-Life (min) | $K_i$ (log(%Inj/ml)-min) | r | p value |
|---|---|---|---|---|
| apoE3;apoER2$^{fl/fl}$ | 1.14 | −0.263 ± 0.04 | 0.88 | <0.0001 |
| apoE3;apoER2$^{\Delta EC}$ | 1.16 | −0.259 ± 0.04 | 0.84 | <0.0001 |
| apoE4;apoER2$^{fl/fl}$ | 1.47 | −0.205 ± 0.04 | 0.77 | <0.0001 |
| apoE4;apoER2$^{\Delta EC}$ | 1.19 | −0.252 ± 0.08 | 0.67 | 0.0116 |

sizes are listed in Table 2. Heat maps visually depicting the variability of the transport rates in each region for each group are presented in Fig 4. Overall, $^{125}$I-insulin BBB transport was more predominantly affected by apoE isoform and minimally affected by loss of endothelial apoER2.

Multiple-time linear regression was used to calculate the transport rate ($K_i$) for the Delta B/S values over time for each region. The y-intercept of each linear regression represented the level of vascular binding for $^{125}$I-insulin ($V_i$). 'r' reflects the correlation coefficient. n/s = non-significant linear regression, indicating the rate of transport ($K_i$) could not be calculated. n = sample size

For groups with significant regional $^{125}$I-insulin uptake, we further measured the level of $^{125}$I-insulin vascular binding ($V_i$) in each brain region (Fig 5). There was no difference in whole brain vascular binding due to apoE isoform or loss of endothelial apoER2 (Fig 5). There was a decrease in $^{125}$I-insulin vascular binding in the midbrain in apoE4 compared to apoE3 in apoER2$^{fl/fl}$ mice (p = 0.0498) (Fig 5K). There was a significant increase in the level of $^{125}$I-insulin vascular binding in the pons/medulla due to loss of endothelial apoER2 in apoE4 mice (p = 0.039) (Fig 5L).

## Discussion

We investigated whether endothelial cell apoER2 modulates the effects of apoE3 or apoE4 on insulin BBB transport. We found that regional insulin BBB transport in the olfactory bulb, midbrain, frontal cortex, parietal cortex, occipital cortex, midbrain, and pons are affected by apoE isoform in mice fed a Western diet, similar to our previous findings [15]. We determined that the presence versus absence of endothelial apoER2 affects BBB insulin transport in the setting of apoE3 in the frontal cortex and pons/medulla. The present data support that insulin BBB transport is regionally regulated and indicate transport is impacted by apoE isoform and largely unaffected by apoER2 in the presence of a Western diet.

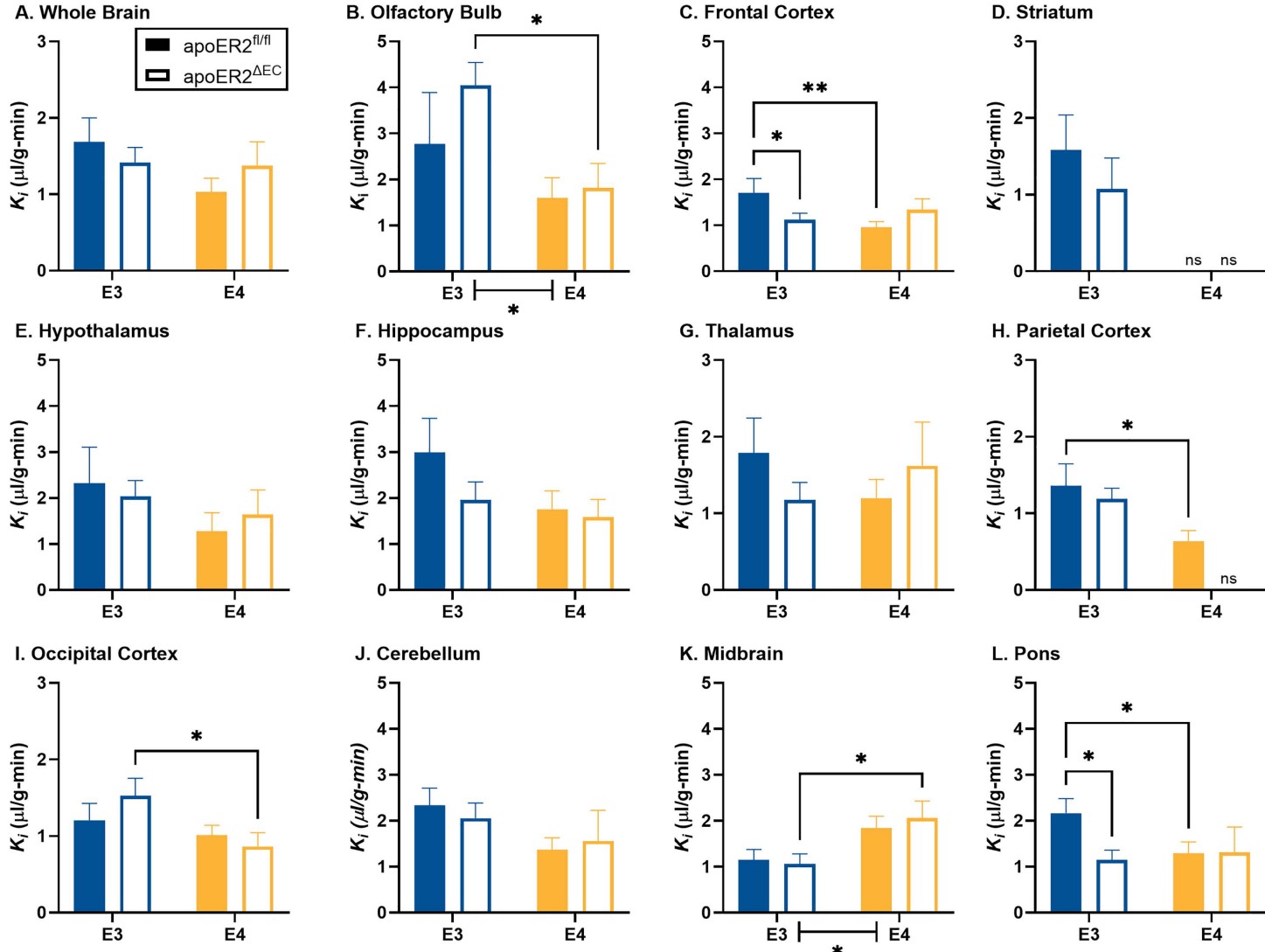

**Fig 3. Brain regional delta linear $^{125}$I-insulin BBB transport rates.** The effects of apoE isoform and endothelial apoER2 on $^{125}$I-insulin BBB transport are graphed for A) whole brain, B) olfactory bulb, C) frontal cortex, D) striatum, E) hypothalamus, F) hippocampus, G) thalamus, H) parietal cortex, I) occipital cortex, J) cerebellum, K) midbrain, and L) pons/medulla. Means reported ± SEM. ANOVA differences are indicated by brackets between E3 vs E4. Fisher's LSD post hoc differences are marked, *p < 0.05, **p < 0.01. Student's t test, *p < 0.05, used for D) striatum and H) parietal cortex. Sample sizes are listed in Table 2 with multiple-time linear regression analysis. ns = linear regression was not statistically significant, and the $K_i$ could not be calculated.

Regarding the effects of apoE4, it is informative to have determined that its detrimental actions on insulin BBB transport are not mediated by endothelial apoER2. A role for another apoE receptor or other mechanism needs to be considered.

We assessed whether brain region vascular space is affected by endothelial apoER2. The loss of endothelial apoER2 had minimal effect on vascular space, only causing an increase in the parietal cortex in apoE4 mice. That localized finding may be explained by the loss of Reelin-apoER2 promotion of endothelial vasoconstriction due to the inhibition of endothelial NO synthase [32]. However, since vascular space was not altered elsewhere, there is likely minimal if any role for Reelin in the observations made with endothelial apoER2 silencing. Using our methods, alterations in cerebrovascular

Table 2. Regional $^{125}$I-insulin pharmacokinetics.

| Region | apoE3;apoER2$^{fl/fl}$ | | | | apoE3;apoER2$^{\Delta EC}$ | | | | apoE4;apoER2$^{fl/fl}$ | | | | apoE4;apoER2$^{\Delta EC}$ | | | |
|---|---|---|---|---|---|---|---|---|---|---|---|---|---|---|---|---|
| | $K_i$ (µL/g-min) | r | $V_i$ (µL/g) | n | $K_i$ (µL/g-min) | r | $V_i$ (µL/g) | n | $K_i$ (µL/g-min) | r | $V_i$ (µL/g) | n | $K_i$ (µL/g-min) | r | $V_i$ (µL/g) | n |
| Whole Brain | 1.69±0.31 | 0.81 | 5.8±1.2 | 17 | 1.42±0.20 | 0.84 | 7.0±0.8 | 23 | 1.03±0.18 | 0.79 | 4.6±0.9 | 22 | 1.38±0.31 | 0.83 | 5.8±1.4 | 11 |
| Olfactory Bulb | 2.77±1.12 | 0.56 | 6.5±3.8 | 16 | 4.05±0.50 | 0.86 | 5.6±2.7 | 25 | 1.60±0.44 | 0.61 | 5.6±2.5 | 24 | 1.82±0.53 | 0.75 | 3.8±2.8 | 11 |
| Frontal Cortex | 1.71±0.31 | 0.83 | 4.3±1.1 | 16 | 1.12±0.14 | 0.86 | 6.1±0.7 | 24 | 0.96±0.13 | 0.86 | 3.7±0.6 | 22 | 1.34±0.24 | 0.86 | 3.6±1.0 | 11 |
| Striatum | 1.58±0.46 | 0.65 | 2.1±2.0 | 18 | 1.08±0.40 | 0.49 | 2.9±2.2 | 25 | 0.003±0.25 (n/s) | 0 | 6.2±1.4 | 24 | 0.47±0.44 (n/s) | 0.32 | 3.5±2.2 | 12 |
| Hypo-thalamus | 2.32±0.78 | 0.65 | 3.3±2.6 | 14 | 2.04±0.35 | 0.79 | 4.7±1.8 | 23 | 1.28±0.40 | 0.6 | 3.7±1.8 | 20 | 1.64±0.54 | 0.7 | 2.3±2.7 | 12 |
| Hippo-campus | 2.99±0.75 | 0.73 | 5.5±2.6 | 18 | 1.96±0.39 | 0.73 | 6.4±1.7 | 23 | 1.75±0.41 | 0.71 | 2.6±1.7 | 20 | 1.58±0.38 | 0.81 | 4.6±1.7 | 11 |
| Thalamus | 1.79±0.46 | 0.7 | 5.3±2.0 | 18 | 1.18±0.23 | 0.75 | 8.2±1.1 | 23 | 1.20±0.25 | 0.73 | 5.1±1.4 | 23 | 1.62±0.57 | 0.69 | 6.7±2.5 | 11 |
| P. Cortex | 1.36±0.29 | 0.77 | 5.0±1.1 | 17 | 1.19±0.14 | 0.88 | 5.3±0.7 | 24 | 0.64±0.14 | 0.71 | 4.6±0.8 | 23 | 0.53±0.39 (n/s) | 0.39 | 6.5±2.0 | 12 |
| O. Cortex | 1.20±0.23 | 0.81 | 5.2±1.0 | 17 | 1.53±0.23 | 0.83 | 6.2±1.0 | 24 | 1.01±0.13 | 0.71 | 4.4±0.7 | 22 | 0.86±0.18 | 0.88 | 6.9±0.8 | 10 |
| Cerebel-lum | 2.33±0.38 | 0.84 | 8.1±1.6 | 18 | 2.06±0.33 | 0.81 | 9.4±1.4 | 23 | 1.37±0.26 | 0.74 | 6.0±1.5 | 24 | 1.56±0.67 | 0.62 | 10.3±2.9 | 11 |
| Midbrain | 1.15±0.23 | 0.79 | 5.3±1.0 | 18 | 1.06±0.22 | 0.73 | 6.3±0.9 | 23 | 1.85±0.25 | 0.91 | 2.4±0.7 | 14 | 2.07±0.37 | 0.8 | 4.0±1.5 | 10 |
| Pons | 2.17±0.32 | 0.79 | 6.5±1.2 | 17 | 1.15±0.21 | 0.73 | 9.2±1.1 | 22 | 1.29±0.25 | 0.91 | 4.6±1.2 | 21 | 1.31±0.55 | 0.62 | 9.2±2.7 | 11 |

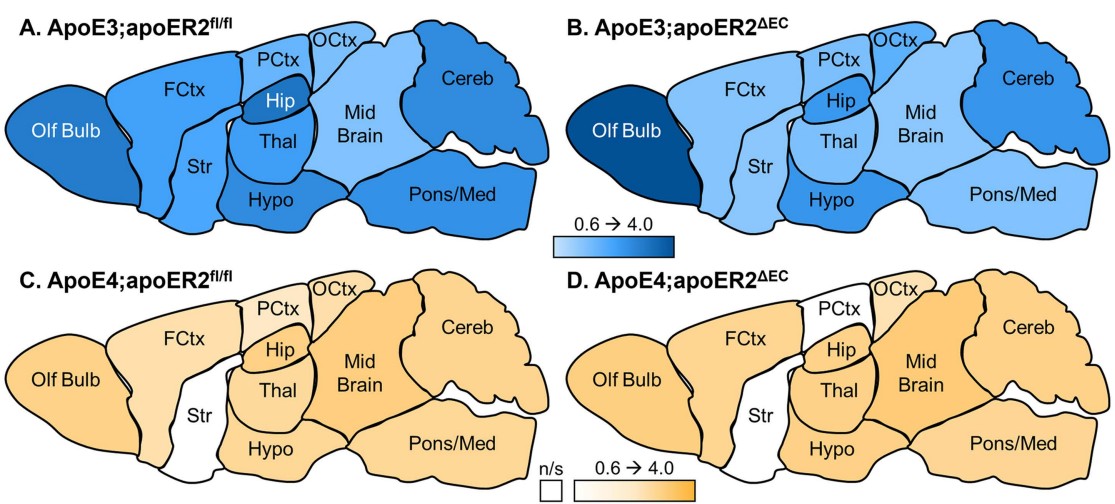

**Fig 4. Regional transport rates (µl/g-min) as represented by a heat map.** Heat maps were generated based on the transport rates listed in Table 2. The $K_i$ for each group was color coded based on the rate, within a range of 0.6-4.0 µl/g-min. White indicates non-significant (n/s) transport. Olf bulb- olfactory bulb, FCtx- frontal cortex, Str- striatum, Hypo- hypothalamus, Thal- thalamus, Hip- hippocampus, PCtx- parietal cortex, OCtx- occipital cortex, Cereb- cerebellum, Pons/Med- pons/medulla.

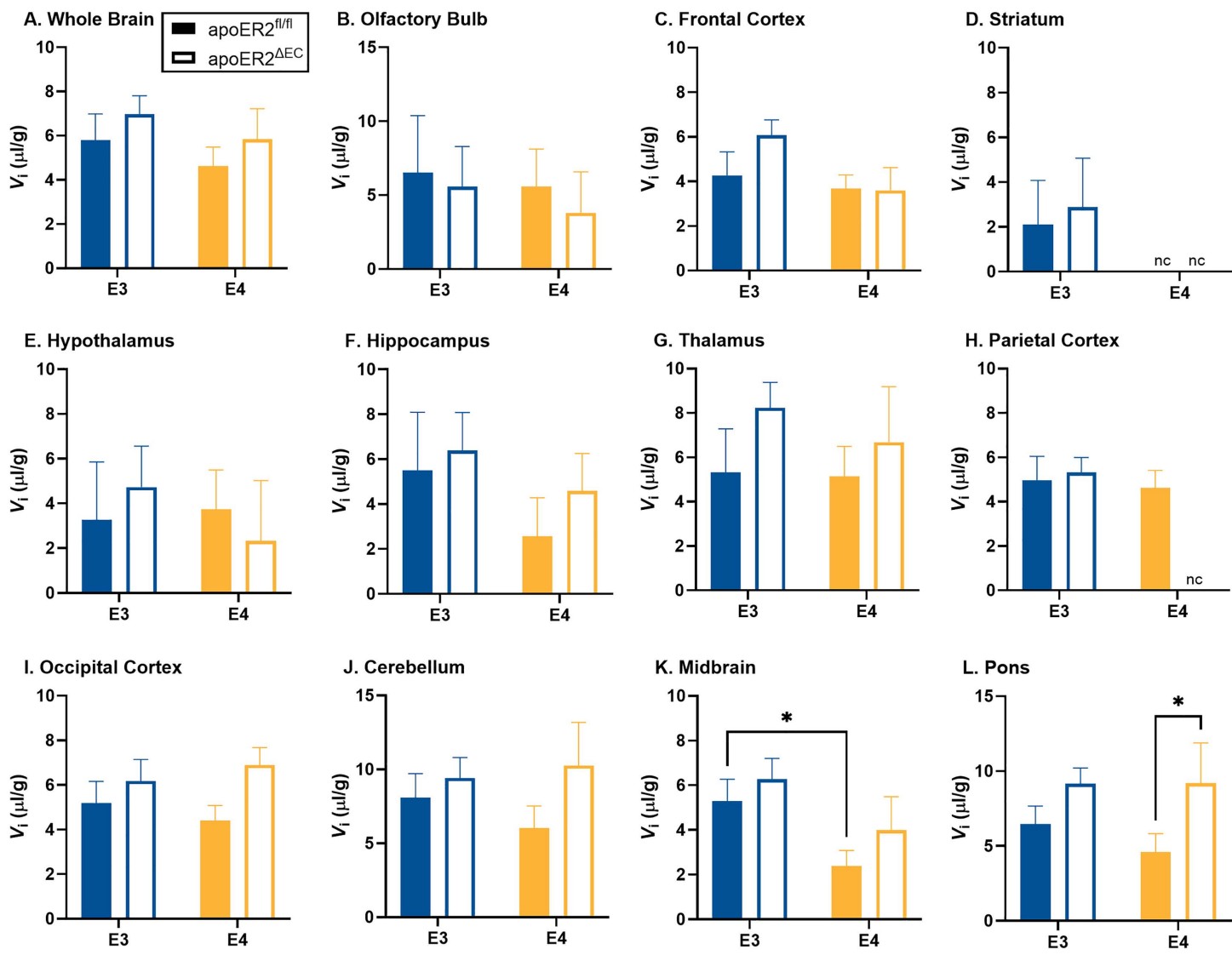

**Fig 5. Brain regional $^{125}$I-insulin vascular binding.** The effects of apoE isoform and endothelial apoER2 on $^{125}$I-insulin BBB vascular binding are graphed for A) whole brain, B) olfactory bulb, C) frontal cortex, D) striatum, E) hypothalamus, F) hippocampus, G) thalamus, H) parietal cortex, I) occipital cortex, J) cerebellum, K) midbrain, and L) pons/medulla. Means reported ± SEM. Fisher's LSD post hoc differences are marked, *p < 0.05, **p < 0.01. Student's t test used for D) striatum and H) parietal cortex. Sample sizes are listed in Table 2 with multiple-time linear regression analysis. nc = not calculated as the linear regression was not statistically significant, and the $V_i$ could not be calculated.

space could indicate differences in modifications of BBB function (i.e., the diameter of the blood vessels) or overall differences in BBB structure (i.e., vascularization). We have previously reported vascular space is modified by apoE4 isoform in a diet and sex specific manner [15], but the current study only investigated the effects in male mice on a Western diet. Levels of $^{125}$I-insulin reaching each brain region were corrected for brain weight and vascular space so these differences do not affect our pharmacokinetic analysis. Serum clearance of insulin was unaffected by the presence or absence of endothelial apoER2.

Statistically significant relations existed between the delta brain region/serum ratios and exposure time for all regions in all mouse groups except the striatum in apoE4;apoER2$^{fl/fl}$ and apoE4;apoER2$^{\Delta EC}$ mice and the parietal cortex in

apoE4;apoER2$^{\Delta EC}$ mice. The transport rates calculated from these relations were compared due to loss of the endothelial cell apoER2 in each region. There was no statistical difference in the insulin transport rate for the whole brain. The regional decreases in the transport rate in apoE4;apoER2$^{fl/fl}$ mice compared to apoE3;apoER2$^{fl/fl}$ mice recapitulates what has been observed before in apoE3 vs apoE4 mice fed a Western diet [15].

Although there were limited regional changes in insulin BBB transport reliant on endothelial cell apoER2, there was a significant reduction in the transport rate within the frontal cortex and pons/medulla only in apoE3 mice. The frontal cortex is a region important in memory. The pons/medulla is a part of the brainstem, connecting the brain to the spinal cord and sub-regions within this brain region are implicated in satiety and nausea. Using public databases, we identified the apoER2 gene (*Irp8*) is most predominantly expressed in brain endothelial cells [7], specifically located in the capillaries compared to veins or arteries [33]. Regionally, expression is greatest in the olfactory bulb, cerebellum, and hippocampus and least in the pons medulla and hypothalamus, with no expression detected in the striatum [34]. Since we observed minor differences in insulin BBB transport due to loss of endothelial apoER2, we do not believe the expression pattern of apoER2 is involved.

In addition to transport rates, we calculated the initial level of vascular binding for insulin at the BBB. Changes in insulin BBB binding could alter the intracellular signaling within the brain endothelial cell or impact insulin signaling within that region [35]. In apoE3 mice, a loss of the endothelial cell apoER2 did not impact insulin BBB binding. In apoE4 mice, there was a significant increase in the level of vascular binding for insulin at the BBB selectively in the occipital cortex. These data indicate endothelial cell apoER2 may play a limited role in regulating brain endothelial cell insulin binding in this region.

In this study, we demonstrated that a loss of a primary receptor in endothelial cells for apoE has a minimum impact on insulin BBB transport. How the specific isoform of a ligand, apoE, could exert a discrete effect on insulin BBB transport in the absence of its receptor, apoER2, remains unclear. ApoE is known to bind to other receptors or binding proteins in endothelium, including LRP1, LDLR, and heparan sulfate proteoglycans [36–38]. Despite the focus on the pharmacokinetic of insulin BBB transport, involvement of these receptors in mediating insulin BBB transport warrants further molecular investigation, focusing on isolated brain microvessel-enriched extracts or spatial immunohistochemistry tools to explore changes in endothelial signaling. In addition, apoE isoform is known to modify its own lipidation status, with E4 being more poorly lipidated compared to apoE3 [39]. BBB function, such as MFSD2A or LRP1 receptor binding which can induce transporter or transcytosis activity, is affected by lipids, lipoproteins, and apolipoproteins [40], and lipidation status of apoE may differentially modulate insulin transport across BBB. These observations indicate apoE isoform can still influence insulin BBB transport in the absence of endothelial apoER2. It should be noted that the current work entailed studies of human apoE3 and apoE4 actions mediated by mouse apoER2. Although it is known that apoE3 and apoE4 have comparable binding to human apoER2 [41], and apoE3 and apoE4 actions via mouse apoER2 have previously been interrogated in neurons [42], the binding affinity and functional interactions of human apoE3 and apoE4 with mouse apoER2 have not yet been characterized.

In summary, these data confirm that apoE4 predominantly blunts insulin BBB transport, and they support a minor regional role, particularly in the frontal cortex and pons/medulla, of endothelial cell apoER2 in modulating insulin BBB transport dependent on apoE isoform. These data give us further insight into how insulin BBB transport may be regulated. Notably, these findings are specific to transport in obese mice fed a Western diet, when insulin transport is already reduced compared to those on a control diet [14,15]. BBB transport is heterogeneous throughout the whole brain, requiring further investigation into the regional regulation of BBB function. These results indicate that the known regional variation in insulin BBB transport is not largely regulated by the endothelial cell apoER2.

## Supporting information

**S1 Fig. Brain regional weight.** The effects of apoE isoform and endothelial apoER2 on brain region weight are graphed for A) whole brain, B) olfactory bulb, C) frontal cortex, D) striatum, E) hypothalamus, F) hippocampus, G) thalamus, H) parietal cortex, I) occipital cortex, J) cerebellum, K) midbrain, and L) pons/medulla. Means reported±SEM. ANOVA differences are indicated by brackets between E3 vs E4. Fisher's LSD post hoc differences are marked, *p<0.05, **p<0.01. Final sample

sizes reflect apoE3;apoER2$^{fl/fl}$ n = 18, apoE3;apoER2$^{\Delta EC}$ n = 27, apoE4;apoER2$^{fl/fl}$ n = 23, apoE4;apoER2$^{\Delta EC}$ n = 12 for all regions except when outliers were removed by the ROUT method (Q = 1%) and included OB: n = 1 apoE4;apoER2$^{fl/fl}$; Pons: n = 1 apoE3;apoER2$^{fl/fl}$, n = 5 apoE3;apoER2$^{\Delta EC}$. WB: whole brain, OB: olfactory bulb, FCtx: frontal cortex, Str: striatum, Thal: thalamus, Hippo: hippocampus, Hypo: hypothalamus, PCtx: parietal cortex, OCtx: occipital cortex, CB: cerebellum. (TIF)

**S2 Fig. Vascular space over time within each brain region.** Multiple-time linear regressions for $^{99m}$Tc-albumin brain/serum (B/S) ratios were plotted against exposure time for each brain region (A-L). There was no significant linear regression within each region for any group, except in the G) thalamus in which the apoE4;apoER2$^{fl/fl}$ group had a positive linear regression (r = 0.47, p = 0.026). Final sample sizes reflect apoE3;apoER2$^{fl/fl}$ n = 17, apoE3;apoER2$^{\Delta EC}$ n = 25, apoE4;apoER2$^{fl/fl}$ n = 24, apoE4;apoER2$^{\Delta EC}$ n = 11 for most regions with additional outliers removed by the ROUT method (Q = 1%) including, Frontal Cortex: n = 1 apoE3;apoER2$^{\Delta EC}$ (n = 24 total), Thalamus: n = 1 apoE3;apoER2$^{\Delta EC}$ n = 24 total), and n = 2 apoE4;apoER2$^{fl/fl}$ (n = 22 total), Occipital Cortex: n = 2 apoE3;apoER2$^{\Delta EC}$ (n = 23 total) and n = 1 apoE4;apoER2$^{fl/fl}$ (n = 10 total), Pons: n = 1 apoE3;apoER2$^{fl/fl}$ (n = 16 total). (TIF)

**S3 Fig. Delta linear $^{125}$I-insulin BBB transport over time within each brain region.** Multiple-time linear regressions for delta brain/serum (B/S) ratios were plotted against exposure time for each brain region (A-L). There was significant linear regression within each region for most groups (solid or dotted lines), except in the D) striatum in which the apoE4;apoER2$^{fl/fl}$ group (r = 0, p = 0.99) and apoE4;apoER2$^{\Delta EC}$ group (r = 0.32, p = 0.31) did not have a significant linear regression (r = 0.39, p = 0.21), nor the H) parietal cortex in the apoE4;apoER2$^{\Delta EC}$ group (r = 0.39, p = 0.21). (TIF)

## Author contributions

**Conceptualization:** Chieko Mineo, Philip W Shaul, Elizabeth M Rhea.

**Data curation:** Peter Thomas, Kim Hansen, Elizabeth M Rhea.

**Formal analysis:** Van Nguyen, Riley Weaver, William A Banks, Elizabeth M Rhea.

**Funding acquisition:** Chieko Mineo, Philip W Shaul, Elizabeth M Rhea.

**Investigation:** Peter Thomas, Riley Weaver, Chieko Mineo, Philip W Shaul, Elizabeth M Rhea.

**Methodology:** Anastasia Sacharidou, Elizabeth M Rhea.

**Project administration:** Elizabeth M Rhea.

**Resources:** Elizabeth M Rhea.

**Supervision:** Chieko Mineo, Philip W Shaul, Elizabeth M Rhea.

**Validation:** Elizabeth M Rhea.

**Visualization:** Chieko Mineo, Philip W Shaul, Elizabeth M Rhea.

**Writing – original draft:** William A Banks, Chieko Mineo, Philip W Shaul, Elizabeth M Rhea.

**Writing – review & editing:** Peter Thomas, Van Nguyen, Riley Weaver, Kim Hansen, Anastasia Sacharidou, William A Banks, Chieko Mineo, Philip W Shaul, Elizabeth M Rhea.

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
