## [Decision Letter · Decision Letter 0]

18 Nov 2025

Dear Dr. Rhea,

Thank you for submitting your manuscript to PLOS ONE. After careful consideration, we feel that it has merit but does not fully meet PLOS ONE’s publication criteria as it currently stands. Therefore, we invite you to submit a revised version of the manuscript that addresses the points raised during the review process.

We look forward to receiving your revised manuscript.

Kind regards,

Sungho Maeng, M.D., Ph.D.

Academic Editor

PLOS ONE

Journal Requirements:

“The Veterans Affairs Research and Development (EMR and WAB) NIH R01HL144969 (PWS), R01HL144969-S1 (PWS), and NIH R01HD094395 (CM) helped support this work”

4. Please note that funding information should not appear in the Acknowledgments section or other areas of your manuscript. We will only publish funding information present in the Funding Statement section of the online submission form. Please remove any funding-related text from the manuscript.

Reviewers' comments:

Reviewer's Responses to Questions

**Comments to the Author**

1. Is the manuscript technically sound, and do the data support the conclusions?

Reviewer #1: Yes

Reviewer #2: Yes

2. Has the statistical analysis been performed appropriately and rigorously?

Reviewer #1: Yes

Reviewer #2: Yes

3. Have the authors made all data underlying the findings in their manuscript fully available?

Reviewer #1: Yes

Reviewer #2: No

4. Is the manuscript presented in an intelligible fashion and written in standard English?

Reviewer #1: Yes

Reviewer #2: Yes

Reviewer #1: This is a review of the manuscript entitled “Role of the endothelial cell apolipoprotein E receptor 2 in modulating the effects of apoE3 and apoE4 on insulin blood-brain barrier transport” by Thomas et al. The authors investigate whether the murine apolipoprotein E receptor 2 (ApoER2) expressed in brain endothelial cells contributes to insulin transport across the blood-brain barrier (BBB), and whether this process is modulated by human apoE isoforms, apoE3 and apoE4. Using endothelial-specific ApoER2 knockout mice crossed with humanized apoE3 or apoE4 lines, the authors report minor regional differences in insulin BBB transport in the context of apoE3, but find that apoE4’s effects on insulin transport appear largely independent of endothelial Apoer2.

The study is well designed and addresses an important question regarding the role of apoE isoforms and endothelial ApoER2 in insulin transport across the BBB. The data are generally well interpreted, and the use of radiolabeled tracers is appropriate. However, several issues, particularly regarding data presentation, validation of the knockout, and interpretation of regional effects, should be addressed to improve clarity and strengthen the conclusions.

Major comments:

1. The presentation of animal numbers in the figure legends and graphs is confusing. While the text states that n = 4 animals were used per group, the figures instead emphasize the number of outliers removed (e.g., “n = 1 outlier”), rather than clearly indicating the number of animals included in the analysis. I recommend that the authors explicitly indicate the number of animals analyzed for each group in the figure or legend, ideally by showing the actual data points (e.g., 2, 3, or 4 points per bar). This would greatly improve clarity and transparency. Additionally, it would be helpful to know whether the same animal was removed as an outlier across multiple brain regions, or whether different animals were excluded for different regions. This detail could inform the reader about potential variability or technical issues in the dataset.

2. The authors rely on a previously published study (line 74) to support effective endothelial-specific deletion of ApoER2 using VE-cadherin-Cre; however, no confirmation of receptor loss in endothelial cells is provided in the current study. Given that the key finding is a lack of effect on insulin transport, it would strengthen the conclusions to validate that Apoer2 was indeed deleted in the animals analyzed. While I recognize that tissues used for radiotracer quantification may not be suitable for additional analyses (e.g., IHC or qPCR), the authors could consider including parallel validation data from separate animals, or referencing relevant confirmatory data (e.g., mRNA, protein, or reporter expression) if available. Without such confirmation, the possibility of incomplete recombination or variable Cre efficiency in this cohort remains a caveat and should be acknowledged.

3. The results highlight regional differences in insulin transport across the BBB, particularly in apoE3-expressing mice. However, the discussion does not address whether regional variation in ApoER2 expression might underlie or contribute to these differences. Incorporating existing expression data, such as from the Allen Brain Atlas, would strengthen the interpretation of the findings and provide valuable context regarding the receptor’s distribution in the brain.

4. The study examines the effects of human apoE isoforms in a mouse model, yet there is no discussion of how human apoE variants interact with mouse ApoER2, or whether these interactions differ across isoforms. It would be important to clarify whether the binding affinity and functional interactions of human apoE with mouse ApoER2 have been characterized, and how this might affect the interpretation of isoform-specific effects on insulin transport.

Minor comments:

5. Line 70 – The manuscript should specify which humanized apoE mouse line was used, as multiple lines are now available. Providing the exact model (including source or reference) would improve clarity and reproducibility.

6. Line 151 – Radiolabeled albumin is referred to only as a "vascular marker," but its function as a negative control for BBB permeability should be made explicit. Since albumin does not normally cross the BBB, it serves to distinguish active transport (e.g., insulin) from passive leakage or vascular entrapment. This clarification would enhance the methodological transparency.

7. If mice were purchased from Jackson Labs or other vendors, the corresponding strain names and stock numbers should be listed in the methods section for clarity and reproducibility.

8. Since Reelin-ApoER2 signaling has been associated with endothelial vasodilation, the authors should consider measuring Reelin protein levels, ApoER2 expression (as a positive control), and phospho-Dab1 levels in endothelial cells to explore this potential mechanism further.

9. To evaluate endothelial signaling, the authors could isolate microvessel-enriched brain extracts for Western blot, or alternatively use RNAscope or immunohistochemistry for cell-specific analysis.

10. It would also be informative to measure circulating ApoE levels across the four mouse lines. This could be done most quantitatively by ELISA, to determine whether differences in plasma ApoE might contribute to the observed effects.

Reviewer #2: The use of constitutive Cdh5-Cre might be problematic, as it also deletes APOE in hematopoietic stem cells, thereby altering multiple downstream immune lineages. This confounds the interpretation of endothelial-specific effects on the insulin BBB transport through ApoER2. The authors should at least discuss this limitation.

ApoER2 is only one of several ApoE receptors expressed by endothelial cells. The manuscript does not evaluate potential compensatory upregulation of other lipoprotein receptors following ApoER2 deletion, which limits the conclusions regarding receptor-specific mechanisms. At least the authors should perform some WB with their brain region samples.

The manuscript does not clearly report the final number of mice included in each analysis. It is difficult to assess whether the study is adequately powered without these details.

The rationale linking the established ApoE isoform–dependent differences in ApoER2 binding to region-specific brain effects is not clearly explained and remains difficult to interpret. Additional evidence is needed to support this connection. For instance, do ApoER2 protein levels vary across the implicated brain regions, or are there regional differences in ApoE abundance?

**Do you want your identity to be public for this peer review?** For information about this choice, including consent withdrawal, please see our Privacy Policy

Reviewer #1: **Yes:** Uwe Beffert

Reviewer #2: No

---

## [Author Response · Author response to Decision Letter 1]

22 Dec 2025

Response to Reviewers

PONE-D-25-44397

Author Response: We thank the reviewers for their thoughtful critiques. We have responded below to each comment and made note of our revisions in the resubmitted version. All changes are tracked in our resubmission so the Reviewers can accurately identify the newly added/edited text. Thank you for your reconsideration of our revised manuscript.

Reviewer #1: This is a review of the manuscript entitled “Role of the endothelial cell apolipoprotein E receptor 2 in modulating the effects of apoE3 and apoE4 on insulin blood-brain barrier transport” by Thomas et al. The authors investigate whether the murine apolipoprotein E receptor 2 (ApoER2) expressed in brain endothelial cells contributes to insulin transport across the blood-brain barrier (BBB), and whether this process is modulated by human apoE isoforms, apoE3 and apoE4. Using endothelial-specific ApoER2 knockout mice crossed with humanized apoE3 or apoE4 lines, the authors report minor regional differences in insulin BBB transport in the context of apoE3, but find that apoE4’s effects on insulin transport appear largely independent of endothelial Apoer2.

The study is well designed and addresses an important question regarding the role of apoE isoforms and endothelial ApoER2 in insulin transport across the BBB. The data are generally well interpreted, and the use of radiolabeled tracers is appropriate. However, several issues, particularly regarding data presentation, validation of the knockout, and interpretation of regional effects, should be addressed to improve clarity and strengthen the conclusions.

Major comments:

1. The presentation of animal numbers in the figure legends and graphs is confusing. While the text states that n = 4 animals were used per group, the figures instead emphasize the number of outliers removed (e.g., “n = 1 outlier”), rather than clearly indicating the number of animals included in the analysis. I recommend that the authors explicitly indicate the number of animals analyzed for each group in the figure or legend, ideally by showing the actual data points (e.g., 2, 3, or 4 points per bar). This would greatly improve clarity and transparency. Additionally, it would be helpful to know whether the same animal was removed as an outlier across multiple brain regions, or whether different animals were excluded for different regions. This detail could inform the reader about potential variability or technical issues in the dataset.

Author Response: We have revised our Figure legends to more clearly list the sample sizes, which are comprised of apoE3;apoER2fl/fl n = 17, apoE3;apoER2ΔEC n = 25, apoE4;apoER2fl/fl n = 24, apoE4;apoER2ΔEC n = 11. As there are 10 brain regions for 4 genotypes represented in each figure, we have listed the starting sample size number and then included specific outliers for regions/groups, allowing the reader to determine the final number of samples included for each mouse genotype and region. Sample sizes represented in the pharmacokinetic data (Figure 3) are listed in Table 2. We have included details about whether the outliers represented the same animal across brain regions in the Statistics methods section. We decided not to change the figures to show the individual data points due to the large sample size for some of the groups.

2. The authors rely on a previously published study (line 74) to support effective endothelial-specific deletion of ApoER2 using VE-cadherin-Cre; however, no confirmation of receptor loss in endothelial cells is provided in the current study. Given that the key finding is a lack of effect on insulin transport, it would strengthen the conclusions to validate that Apoer2 was indeed deleted in the animals analyzed. While I recognize that tissues used for radiotracer quantification may not be suitable for additional analyses (e.g., IHC or qPCR), the authors could consider including parallel validation data from separate animals, or referencing relevant confirmatory data (e.g., mRNA, protein, or reporter expression) if available. Without such confirmation, the possibility of incomplete recombination or variable Cre efficiency in this cohort remains a caveat and should be acknowledged.

Author Response: We have confirmed the endothelial cell deletion of ApoER2 resulting from crossing of ApoER2 floxed mice (which we originally generated) with VE-cadherin-Cre mice in two independent prior publications. One is original reference #24, and the second from earlier this year is PMID: 40907945, which we also now cite. The Methods section has been revised to now mention this thorough demonstration of the loss of endothelial cell ApoER2.

3. The results highlight regional differences in insulin transport across the BBB, particularly in apoE3-expressing mice. However, the discussion does not address whether regional variation in ApoER2 expression might underlie or contribute to these differences. Incorporating existing expression data, such as from the Allen Brain Atlas, would strengthen the interpretation of the findings and provide valuable context regarding the receptor’s distribution in the brain.

Author Response: We appreciate this Reviewer’s thoughtful comment. We have now included a discussion on the regional distribution of ApoER2 alongside the regional variation in insulin BBB transport. Utilizing three independent databases, we identified ApoER2 expression is most abundant in the olfactory bulb, with minimal to no expression in the striatum and pons (Allen Brain Atlas). Additionally, we identified ApoER2 is most abundantly expressed in brain endothelial cells, compared to other CNS cell types (Brain-Seq), and more specifically in the capillaries, where there is prevalent BBB transport (Vine-Seq). However, due to the minimal regional differences of ApoER2 on insulin BBB transport, we do not believe this expression pattern explains the data for insulin BBB transport.

4. The study examines the effects of human apoE isoforms in a mouse model, yet there is no discussion of how human apoE variants interact with mouse ApoER2, or whether these interactions differ across isoforms. It would be important to clarify whether the binding affinity and functional interactions of human apoE with mouse ApoER2 have been characterized, and how this might affect the interpretation of isoform-specific effects on insulin transport.

Author Response: We now mention the possible limitation that human apoE3 and apoE4 actions via mouse ApoER2 were studied with the following text added to the Discussion: It should be noted that the current work entailed studies of human apoE3 and apoE4 actions mediated by mouse ApoER2. Although it is known that apoE3 and apoE4 have comparable binding to human ApoER2 (PMID: 30375977), and apoE3 and apoE4 actions via mouse ApoER2 have previously been interrogated in neurons (PMID: 20547869), the binding affinity and functional interactions of human apoE3 and apoE4 with mouse ApoER2 have not yet been characterized.

Minor comments:

5. Line 70 – The manuscript should specify which humanized apoE mouse line was used, as multiple lines are now available. Providing the exact model (including source or reference) would improve clarity and reproducibility.

Author Response: In our original manuscript, we cited a reference to the original line used, generated by Dr. Maeda in 1999, but have now included the details of this humanized apoE mouse line in the methods section.

6. Line 151 – Radiolabeled albumin is referred to only as a "vascular marker," but its function as a negative control for BBB permeability should be made explicit. Since albumin does not normally cross the BBB, it serves to distinguish active transport (e.g., insulin) from passive leakage or vascular entrapment. This clarification would enhance the methodological transparency.

Author Response: We agree that radiolabeled albumin can also be used as a BBB permeability marker. However, due to the short circulatory time used in our study (<10 min) and lack of brain vascular wash-out, radiolabeled albumin would not detect passive leakage. We have now included a sentence in the Methods to describe why this marker and the accompanying results cannot be used as a marker for BBB integrity.

7. If mice were purchased from Jackson Labs or other vendors, the corresponding strain names and stock numbers should be listed in the methods section for clarity and reproducibility.

Author Response: None of the mice were obtained from vendors, and all their sources are accurately cited.

8. Since Reelin-ApoER2 signaling has been associated with endothelial vasodilation, the authors should consider measuring Reelin protein levels, ApoER2 expression (as a positive control), and phospho-Dab1 levels in endothelial cells to explore this potential mechanism further.

Author Response: In cultured endothelial cells it has been observed that Reelin inhibits endothelial NO synthase via ApoER2 (PMID: 26980442), which would result in relative vasoconstriction. In the studies of vascular space, the loss of endothelial ApoER2 had minimal effect, only causing an increase in vascular space in the parietal cortex in apoE4 mice. That localized finding may be explained by the loss of Reelin-apoER2 promotion of vasoconstriction, but since vascular space was not altered elsewhere, there is likely minimal if any role for Reelin in the observations made with endothelial apoER2 silencing. We therefore respectfully believe that the quantification of Reelin and phospho-Dab1 would add little if anything to this report.

9. To evaluate endothelial signaling, the authors could isolate microvessel-enriched brain extracts for Western blot, or alternatively use RNAscope or immunohistochemistry for cell-specific analysis.

Author Response: We agree these tools would be an exciting, additional component to this story investigating the role of endothelial ApoER2 in insulin BBB transport. As this initial paper describes the transport pharmacokinetics, we have listed these additional studies as future directions in the Discussion.

10. It would also be informative to measure circulating ApoE levels across the four mouse lines. This could be done most quantitatively by ELISA, to determine whether differences in plasma ApoE might contribute to the observed effects.

Author Response: Plasma ApoE has already been compared in humanized apoE3 versus apoE4 mice, both on standard diets and western diets, and the levels of apoE3 and apoE4 were similar (PMID: 10359567 and PMID: 38058491). There is minimal if any gain by quantifying circulating apoE3 and apoE4 in the present project.

Reviewer #2: The use of constitutive Cdh5-Cre might be problematic, as it also deletes APOE in hematopoietic stem cells, thereby altering multiple downstream immune lineages. This confounds the interpretation of endothelial-specific effects on the insulin BBB transport through ApoER2. The authors should at least discuss this limitation.

ApoER2 is only one of several ApoE receptors expressed by endothelial cells. The manuscript does not evaluate potential compensatory upregulation of other lipoprotein receptors following ApoER2 deletion, which limits the conclusions regarding receptor-specific mechanisms. At least the authors should perform some WB with their brain region samples.

Author Response: In our prior work using the same floxed ApoER2 mouse crossed with constitutive Cdh5-Cre mice we documented effective loss of ApoER2 from endothelial cells and no change in ApoER2 expression in bone marrow-derived myeloid cells (Ref. 24). The lack of ApoER2 change in hematopoietic cells is now mentioned in the Methods.

Regarding the possible involvement of other lipoprotein receptors besides ApoER2, we mention LRP1, LDLR and heparan sulfate proteoglycans, which may warrant further molecular investigation, in the Discussion.

The manuscript does not clearly report the final number of mice included in each analysis. It is difficult to assess whether the study is adequately powered without these details.

Author Response: We have clarified the final sample size in each analysis in the Figure Legends. Please see our Response to Reviewer 1, Point 1.

The rationale linking the established ApoE isoform–dependent differences in ApoER2 binding to region-specific brain effects is not clearly explained and remains difficult to interpret. Additional evidence is needed to support this connection. For instance, do ApoER2 protein levels vary across the implicated brain regions, or are there regional differences in ApoE abundance?

Author Response: Please see our response to Reviewer 1, point 3.

---

## [Decision Letter · Decision Letter 1]

2 Feb 2026

Role of the endothelial cell apolipoprotein E receptor 2 in modulating the effects of apoE3 and apoE4 on insulin blood-brain barrier transport

PONE-D-25-44397R1,

We’re pleased to inform you that your manuscript has been judged scientifically suitable for publication and will be formally accepted for publication once it meets all outstanding technical requirements.

Kind regards,

Sungho Maeng, M.D., Ph.D.

Academic Editor

PLOS One

Additional Editor Comments (optional):

Reviewers' comments:

Reviewer's Responses to Questions

**Comments to the Author**

Reviewer #1: All comments have been addressed

Reviewer #2: All comments have been addressed

2. Is the manuscript technically sound, and do the data support the conclusions?

Reviewer #1: Yes

Reviewer #2: Yes

3. Has the statistical analysis been performed appropriately and rigorously?

Reviewer #1: Yes

Reviewer #2: Yes

4. Have the authors made all data underlying the findings in their manuscript fully available?

Reviewer #1: Yes

Reviewer #2: Yes

5. Is the manuscript presented in an intelligible fashion and written in standard English?

Reviewer #1: Yes

Reviewer #2: Yes

Reviewer #1: The authors have done a reasonable job of addressing both reviewers critiques and the revised manuscript is acceptable for publication.

Reviewer #2: (No Response)

**Do you want your identity to be public for this peer review?** For information about this choice, including consent withdrawal, please see our Privacy Policy

Reviewer #1: No

Reviewer #2: No

---

## [Editor Report · Acceptance letter]

PONE-D-25-44397R1

PLOS One

Dear Dr. Rhea,

I'm pleased to inform you that your manuscript has been deemed suitable for publication in PLOS One. Congratulations! Your manuscript is now being handed over to our production team.

Kind regards,

on behalf of

Dr. Sungho Maeng

Academic Editor

PLOS One